# LLM Pretraining with Continuous Concepts

**Jihoon Tack**[1,2]**, Jack Lanchantin**[1]**, Jane Yu**[1]**, Andrew Cohen**[1]**, Ilia Kulikov**[1]**,
Janice Lan**[1]**, Shibo Hao**[1,3]**, Yuandong Tian**[1]**, Jason Weston**[1]**, Xian Li**[1]
[1]FAIR at Meta, [2]KAIST, [3]UC San Diego

## Abstract

Next token prediction has been the standard training objective used in large language model pretraining. Representations are learned as a result of optimizing for token-level perplexity. We propose Continuous Concept Mixing (CoCoMix), a novel pretraining framework that combines discrete next token prediction with continuous concepts. Specifically, CoCoMix predicts "continuous concepts" learned from a pretrained sparse autoencoder and mixes them into the model's hidden state by interleaving with token hidden representations. Through experiments on multiple benchmarks, including language modeling and downstream reasoning tasks, we show that CoCoMix is more sample efficient and consistently outperforms standard next token prediction, knowledge distillation and inserting pause tokens. We find that combining both concept learning and interleaving in an end-to-end framework is critical to performance gains. Furthermore, CoCoMix enhances interpretability and steerability by allowing direct inspection and modification of the predicted concept, offering a transparent way to guide the model's internal reasoning process.

## 1 Introduction

Recent progress in large language models (LLMs) has revolutionized natural language processing (Brown et al., 2020; Dubey et al., 2024) and thus became a core technology in various real-world applications, such as coding assistants (Roziere et al., 2023), search engines (Xuan-Quy et al., 2023), and personal AI assistants (Gao et al., 2023). Central to these breakthroughs is the simple paradigm of next token prediction, which leverages massive amounts of unlabeled text to uncover rich linguistic patterns (Radford et al., 2018; 2019). However, natural language tokens are often superficial (e.g., function words like "the" or "a"), necessitating substantial training for models to acquire high-level reasoning and conceptual understanding while also hindering their ability to tackle long-horizon tasks such as planning (LeCun, 2022; Bachmann and Nagarajan, 2024).

To this end, recent studies have investigated methods that go beyond token-level signals by leveraging richer information to train models. For instance, some approaches target more expressive prediction objectives, such as predicting multiple tokens at once to better capture semantic relationships (Gloeckle et al., 2024; DeepSeek-AI, 2024), while others augment the input with rich signals, e.g., self-generated thought tokens (Zelikman et al., 2024), or fixed pause tokens (Goyal et al., 2024) prior to next token prediction. Moreover, emerging evidence suggests that LLMs inherently encode high-level concepts and reasoning processes in their latent representations (Deng et al., 2023; Yang et al., 2024), indicating that replacing discrete language tokens with continuous latent representations has promise in improving reasoning efficiency (Hao et al., 2024). While token-level modeling remains important for coherent text generation, the key challenge is to enrich or supplement these language tokens so that LLMs can learn more abstract reasoning abilities and long-range dependencies.

This raises a key question: can we augment next token prediction objective to explicitly model concepts in a latent representation space, thereby bridging semantic abstraction and fine-grained token-level guidance?

To this end, we draw inspiration from recent findings that *Sparse Autoencoders (SAEs)* can effectively isolate meaningful latent features in LLMs by capturing the high-level semantic concepts (Cunningham et al., 2023; Bricken et al., 2023). As SAEs are trained to reconstruct the model's hidden state with sparsity constraints, it encourages focusing on a compact set of concept dimensions (Templeton

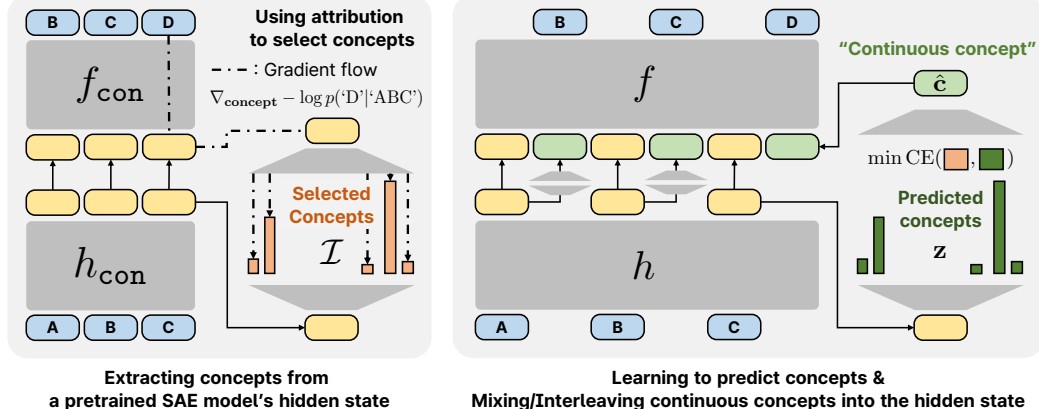

Figure 1: **Overview of CoCoMix.** We use an SAE to extract concepts from a pretrained model's hidden state $h_{con}$ and then select important concepts based on the attribution score (i.e., measuring the influence on the output). These selected concepts are used as labels $\mathcal{I}$ for concept prediction by minimizing the cross-entropy loss $CE(\cdot, \cdot)$. The predicted concepts $\mathbf{z}$ are then compressed into a compact vector, forming a continuous concept $\hat{\mathbf{c}}$, which is mixed into the model's hidden state by interleaving with token hidden representations. Here, the ▨, ▨, and ▨ boxes indicate text tokens, token hidden representation, and continuous concepts, respectively.

et al., 2024). This makes it possible to highlight the pretrained model's concepts—the core semantic directions that underpin model prediction—while avoiding unnecessary features.

We propose Continuous Concept Mixing (CoCoMix), a novel and effective language modeling framework using continuous concepts. Specifically, we extract semantic concepts using a pretrained SAE and select the most influential ones based on attribution scores, which quantify each concept's influence on the model's output. The model is then trained to predict these selected concepts from its hidden state using a cross-entropy loss. Once multiple concepts are predicted, we compress them into a single *continuous concept* and *mix* (or insert) into the hidden states by interleaving with token embeddings, thereby directly contributing to the next token prediction. An additional benefit is that one can probe or analyze the predicted concepts, enabling controllable generation and improving model interpretability. An overview of CoCoMix is shown in Figure 1.

We demonstrate the efficacy of CoCoMix through extensive evaluations across multiple language modeling benchmarks and pretraining model sizes, ranging from million-scale to billion-scale parameter sizes. For instance, when applied to a 1.38B-sized model, CoCoMix achieves comparable performance with the next token prediction with 21.5% fewer training tokens. Moreover, CoCoMix demonstrates substantial improvements in weak-to-strong supervision scenarios, where concepts extracted from a small model can even be used as ground truth labels to supervise the training of a larger model. Finally, we show that the insertion of compressed concept vectors enables us to steer and control the model by probing the predicted concept during generation.

## 2 CoCoMix: Continuous Concept Mixing

In this section, we present Continuous Concept Mixing (CoCoMix), a framework that extends next-token prediction with continuous concepts. We first review our problem setup of interest and the concept extraction process using a sparse autoencoder (SAE) in Section 2.1. Then, we describe the concept selection framework and explain how the model predicts and interleaves these concepts into its hidden state to improve language modeling in Section 2.2.

### 2.1 Prelim: Problem Setup and Sparse Autoencoder

**Problem setup.** We consider a language-modeling task over a vocabulary $\mathcal{V}$, with a training corpus $\mathcal{D} \subseteq \mathcal{V}^*$, where each sequence $\mathbf{x} = (x_1, x_2, \ldots) \in \mathcal{D}$ is drawn i.i.d. from the data distribution. We require a pretrained language model for concept extraction $\mathcal{M}_{con} = f_{con} \circ h_{con}$ which predicts next

token probability $\mathcal{M}_{\text{con}}(x_t \mid x_{<t})$. Here, we extract the continuous concepts $\mathbf{c}$ from the hidden state $h_{\text{con}}$, where $h_{\text{con}}$ outputs the hidden state at a chosen layer of depth $L_{\text{con}}$, and $f_{\text{con}}$ maps that representation to the next token distribution. We aim to train $\mathcal{M}$ to model the same autoregressive factorization $\mathcal{M}(\mathbf{x}) = \prod_{t=1}^{|\mathbf{x}|} \mathcal{M}(x_t \mid x_{<t}, \mathbf{c}_{<t})$, but by augmenting with continuous concepts $\mathbf{c}$.

**Sparse Autoencoder.** We first propose to extract high-level concepts from a pretrained model's hidden state. We use an SAE, which maps the input to a high-dimensional sparse activation, with the objective of reconstructing the original input (Lee et al., 2006). When applied to an LLM, the SAE decomposes the hidden state into multiple dimensions, each of which can be viewed as a distinct concept capturing semantically meaningful features (Yun et al., 2021; Bricken et al., 2023). We consider the TopK SAE (Makhzani and Frey, 2014), which uses a TopK activation to enforce sparsity.

Formally, for a given input sequence $\mathbf{x}$ and corresponding pretrained model's hidden state at position $t$, denoted as $\mathbf{h}_t^{\text{con}} = h_{\text{con}}(\mathbf{x})_t \in \mathbb{R}^{d_{\text{con}}}$, SAE consists of a linear encoder $E : \mathbb{R}^{d_{\text{con}}} \to \mathbb{R}^C$ and a linear decoder $D : \mathbb{R}^C \to \mathbb{R}^{d_{\text{con}}}$, where $C$ is the dimension of the concept space. The reconstruction process of SAE is as:

$$\mathbf{c}_t^{\text{pre}} = E(\mathbf{h}_t^{\text{con}}), \quad \mathbf{c}_t = \text{TopK}(\mathbf{c}_t^{\text{pre}}), \quad \widehat{\mathbf{h}}_t^{\text{con}} = D(\mathbf{c}_t),$$

where $\mathbf{c}_t^{\text{pre}} \in \mathbb{R}^C$ is the pre-activation concept vector, $\text{TopK}(\cdot)$ zeros out all but the largest $K_{\text{SAE}}$ entries, and $\widehat{\mathbf{h}}_t^{\text{con}}$ is the reconstruction. The SAE is trained by minimizing the reconstruction loss: $\|\mathbf{h}_t^{\text{con}} - \widehat{\mathbf{h}}_t^{\text{con}}\|_2^2$. By enforcing TopK sparsity, the SAE isolates the most critical dimensions in $\mathbf{c}$ that explain the pretrained model's features. Here, each activated element of $\mathbf{c}_t$ is interpreted as a concept.

## 2.2 CONTINUOUS CONCEPT MIXING

Now, we describe the core training pipeline of CoCoMix, which consists of a concept selection framework (refer to the left figure in Figure 1) and two training steps to learn and utilize continuous concepts (refer to the right figure in Figure 1). First, we select important concepts using the attribution score, which measures the influence of each concept on the output. Then, we propose predicting the selected concepts from the model's hidden state using cross-entropy loss, allowing the model to implicitly learn which concepts should be encoded as a hidden representation. Finally, we utilize the predicted concepts to create a "continuous concept", which is interleaved in the hidden states, enabling the model to explicitly learn to use both the continuous concepts as well as the token hidden states. Intuitively, the model selectively learns which concepts are useful for the next token prediction and how it should *mix* them with the token representations.

**Target concept selection using attribution.** While the extracted concept $\mathbf{c}_t$ represents the core features of the current input context, it may not directly indicate which concepts matter most for predicting the ground truth next token $x_{t+1}$ (Templeton et al., 2024). To this end, we propose utilizing *attribution*, a method to measure the causal influence of each concept on the output (Baehrens et al., 2010; Sundararajan et al., 2017). Specifically, the attribution score measures the influence based on the local linear approximation of the effect of changing the concept value. In this paper, we use a simple attribution score that measures the influence by multiplying the loss gradient with a given input (Simonyan and Zisserman, 2014; Shrikumar et al., 2016).

Concretely, for a given input $\mathbf{x}$ and corresponding concept $\mathbf{c}_t$, we define the attribution score $\mathbf{a}_t \in \mathbb{R}^C$:

$$\mathbf{a}_t = \mathbf{c}_t \odot \nabla_{\mathbf{c}_t} - \log f_{\text{con}}(x_{t+1} | D(\mathbf{c}_t), \mathbf{h}_{<t}), \tag{1}$$

where $\odot$ denotes element-wise multiplication. Based on the computed attribution score, we select salient concepts and incorporate them into language modeling.

**Predicting the selected concepts.** For a given attribution score $\mathbf{a}_t$, we first select the indices of the concept that have a high score and use these as discrete labels for concept prediction. Let $\mathcal{I} = \{i_1, \ldots, i_{K_{\text{attr}}}\}$ be the set of indices corresponding to the top $K_{\text{attr}}$ values of $\mathbf{a}_t$, and $\mathbf{h}_t = h(\mathbf{x})_t \in \mathbb{R}^d$ be the model's hidden state of the given input $\mathbf{x}$ at the same token position. Then, we learn to predict these labels using a linear prediction head $M$ that outputs logit $\mathbf{z}_t = M(\mathbf{h}_t) \in \mathbb{R}^C$ by minimizing the following cross-entropy loss $\text{CE}(\cdot, \cdot)$ is as follows:

$$\mathcal{L}_{\text{concept}}(\mathbf{a}_t) = \frac{1}{K_{\text{attr}}} \sum_{i \in \mathcal{I}} \text{CE}(\mathbf{z}_t, i). \tag{2}$$

**Mixing continuous concepts with token embeddings.** To encourage the model to internalize the concept more effectively, we propose "mixing" (i.e., interleaving) the predicted concept with the existing token hidden representations. As our model predicts multiple concepts at once, we propose to compress them into a compact vector through a learnable mapping, which we refer to as a continuous concept. This compressed vector is interleaved with token vectors.

Formally, for a given concept prediction logit $\mathbf{z}_t$, we sparsify the logit using $\mathrm{TopK}$ activation to predict the concepts, then compress them into a continuous concept vector $\hat{\mathbf{c}}_t \in \mathbb{R}^d$:

$$\hat{\mathbf{c}}_t \;=\; \mathbf{W}\,\mathrm{TopK}(\mathbf{z}_t) + \mathbf{b}, \tag{3}$$

where $\mathbf{W} \in \mathbb{R}^{d \times C}$ and $\mathbf{b} \in \mathbb{R}^d$ project the TopK-sparse vector to a $d$-dimensional embedding. We then append $\hat{\mathbf{c}}_t$ to the model's hidden sequence $\hat{\mathbf{c}}_t$ as $(\mathbf{h}_1, \hat{\mathbf{c}}_1, \ldots, \mathbf{h}_t, \hat{\mathbf{c}}_t)$, which is fed into the remaining transformer blocks $f$. Note that this design not only improves the model's performance but also offers interpretability and controllability through analyzing and steering the predicted concept $\mathbf{z}_t$, which can be probed or tuned during the model's generation process. Furthermore, by analyzing the weights of the compression layer $\mathbf{W}$, one can identify which concept is useful for the next token prediction.

This approach shares similarities with intervention techniques in the mechanistic interpretability literature, which modify the hidden state by adding a concept vector to the original hidden state (Zou et al., 2023; Wu et al., 2024). However, unlike intervention methods that directly manipulate the hidden state, our approach treats the predicted concept as a separate unit of information by interleaving it in the hidden state. This design allows the model to process the concept independently, leveraging the pretrained model's internal reasoning reflected in the prediction.

**Training objective.** The final training objective for CoCoMix combines the standard next token prediction loss and the concept prediction term with a tunable coefficient $\lambda$ as follows:

$$\sum_{t=1}^{T-1} -\log f\big(x_{t+1} \mid \mathbf{h}_{\leq t}, \hat{\mathbf{c}}_{\leq t}\big) \;+\; \lambda \mathcal{L}_{\mathrm{concept}}(\mathbf{a}_t). \tag{4}$$

## 3 EXPERIMENTS

We provide an empirical evaluation of CoCoMix by investigating the following questions:

- Can CoCoMix improve the performance of next token prediction? (Figure 2 and Figure 3)
- Does CoCoMix show improvement in weak-to-strong supervision setup compared to other knowledge distillation methods? (Table 1 and Figure 4)
- Does CoCoMix introduce model interpretability and steerability? (Figure 5)
- How does each proposed component of CoCoMix contribute to the performance? (Figure 6)

Before answering each question, we outline the experimental protocol (more details in Appendix A).

**Training setup.** We use a pretrained open-source SAE that is trained on the 124M-sized GPT-2 (Gao et al., 2024), which has demonstrated strong effectiveness despite a somewhat small model size (Chaudhary and Geiger, 2024). Here, we consider training CoCoMix with three different numbers of active parameters, including 69M, 386M, and 1.38B (which is an academic-scale pretraining (Allen-Zhu, 2025)), with a context length of 1024. For the analysis and ablation study, we mainly conducted experiments on the 69M model. Note that this activated parameter count includes those in the new layer (i.e., the concept predictor), whereas for other baselines, we match the same number of active parameters by increasing the hidden state dimension size $d$. CoCoMix utilizes fewer FLOPs than Pause token (as discussed in Section 3.3) but more FLOPs than NTP, due to the interleaving of continuous concepts. We use the OpenWebText dataset (Radford et al., 2019) as the pretraining corpus to use the same distribution used to train $\mathcal{M}_{\mathrm{con}}$. All experiments are conducted with 20B training tokens, except for the main experiment in Section 3.1, which uses 200B tokens.

**Baselines.** We consider the standard pretraining next token prediction (NTP) procedure, and the commonly used distillation method, knowledge distillation (KD) (Hinton et al., 2015), which is widely used in pretraining (Gu et al., 2024; Sanh et al., 2019; Team, 2024). We excluded KD baselines that require multiple models (i.e., more than a single teacher model) to be trained (Gu et al., 2024).

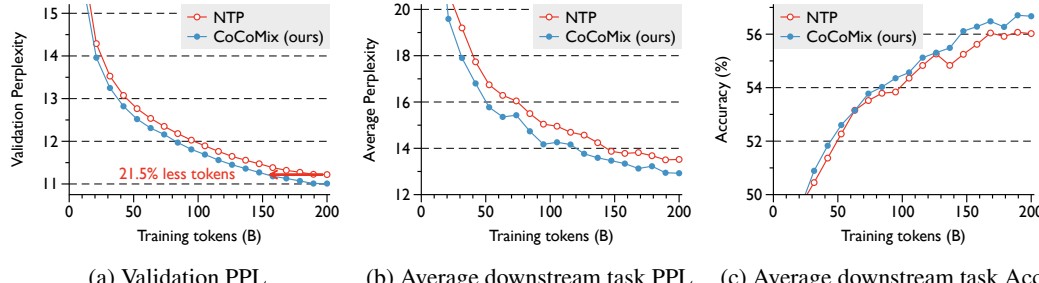

(a) Validation PPL  (b) Average downstream task PPL  (c) Average downstream task Acc

Figure 2: **CoCoMix vs. NTP performance at different training checkpoints.** Each model contains a total of 1.38B parameters. Each model is trained on the OpenWebText dataset. For CoCoMix, the concepts are extracted from a 124M-sized model (10× smaller than the base model). The plots show improvements in: (a) validation perplexity, (b) average perplexity on LAMBADA, WikiText-103, and (c) average accuracy on HellaSwag, PIQA, SIQA, Arc-Easy, and WinoGrande. Here, CoCoMix achieves the same performance of NTP with 21.5% fewer tokens.

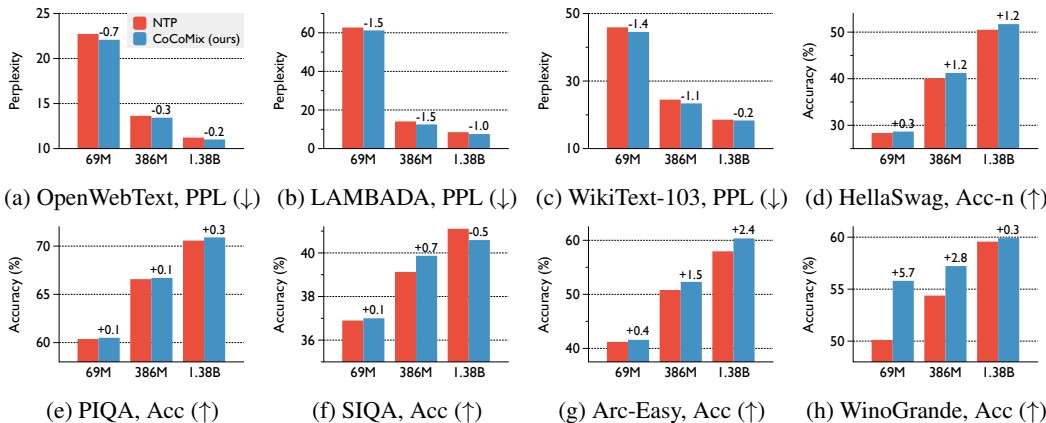

(a) OpenWebText, PPL (↓)  (b) LAMBADA, PPL (↓)  (c) WikiText-103, PPL (↓)  (d) HellaSwag, Acc-n (↑)

(e) PIQA, Acc (↑)  (f) SIQA, Acc (↑)  (g) Arc-Easy, Acc (↑)  (h) WinoGrande, Acc (↑)

Figure 3: **CoCoMix vs. NTP performance at different model sizes.** We consider various model sizes, including 69M, 386M, and 1.38B parameters and train on 200B OpenWebText tokens. We evaluate the models on OpenWebText validation perplexity and downstream datasets LAMBADA, WikiText-103, HellaSwag, PIQA, SIQA, Arc-Easy, and WinoGrande. Overall, CoCoMix achieves stronger performance than NTP in virtually all model–dataset combinations considered.

For KD, we minimize the KL divergence between teacher and student output while balancing the KL term with the NTP loss. Note that using synthetic datasets for distillation (Kim and Rush, 2016; Yu et al., 2024) is excluded due to inefficiency (needs to generate more than 20B tokens).

**Evaluation setup.** For evaluation, we consider the validation perplexity of the pretraining dataset and 7 downstream tasks to benchmark commonsense reasoning and reading comprehension. This includes LAMBADA (Paperno et al., 2016), WikiText-103 (Merity et al., 2017), HellaSwag (Zellers et al., 2019), PIQA (Bisk et al., 2020), Social IQA (SIQA) (Sap et al., 2019), ARC-easy (Clark et al., 2018), and WinoGrande (Sakaguchi et al., 2020) datasets. We also consider OpenWebMath (Paster et al., 2023) as a pretraining dataset to demonstrate that concepts learned from a pretrained LLM can still be applied to CoCoMix, even when the model was trained on a different corpus.

## 3.1 MAIN RESULTS

In this section, we illustrate two core results: i) the comparison with NTP on a relatively large-scale pretraining setup and ii) the comparison with KD baseline, especially on weak-to-strong supervision scenarios where concepts extracted from a small model are used to guide a larger model.

**Improving NTP with CoCoMix at scale.** We present the main result by applying CoCoMix to the NTP by training NTP and CoCoMix on 200B tokens. As shown in Figure 3, CoCoMix consistently and significantly improves the performance of overall downstream tasks on various model sizes.

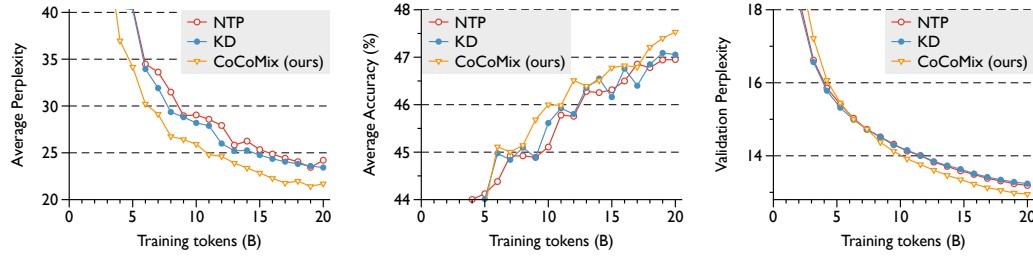

(a) Weak-to-strong: Avg. PPL      (b) Weak-to-strong: Avg. Acc      (c) Distribution shift: Math domain

Figure 4: **CoCoMix vs. Knowledge Distillation (KD).** For our weak-to-strong supervision setup, we train a 386M model where the teacher of KD (or concept extraction for CoCoMix) is a 124M-sized model: we report (a) average perplexity over OpenWebText, LAMABADA, and WikiText and (b) average accuracy over HellaSwag, PIQA, SIQA, Arc-Easy, and WiniGrande dataset. For (c) a distribution shift setup, we train all methods on OpenWebMath, a math specific pretraining corpus.

Table 1: **CoCoMix vs. Next Token Prediction (NTP) vs. Knowledge Distillation (KD).** We report performance on the OpenWebText (OWT) training set, as well as downstream tasks, including LAMBADA (LAMB), WikiText-103 (Wiki), HellaSwag (HellaS), PIQA, Social Interaction QA (SIQA), Arc-Easy (Arc-E), and WinoGrande (WinoG). We train three different sizes of models where 124M model is used as a teacher. All models are trained on 20B tokens sampled from the OpenWebText dataset. The bold indicates the best result.

| Method | Total Params | OWT PPL ($\downarrow$) | LAMB PPL ($\downarrow$) | Wiki PPL ($\downarrow$) | HellaS Acc-n ($\uparrow$) | PIQA Acc ($\uparrow$) | SIQA Acc ($\uparrow$) | Arc-E Acc ($\uparrow$) | WinoG Acc ($\uparrow$) | Avg PPL ($\downarrow$) | Avg Acc ($\uparrow$) |
|---|---|---|---|---|---|---|---|---|---|---|---|
| NTP | 69M | 25.3 | 107.6 | 52.3 | 27.4 | 59.4 | 36.6 | 39.7 | 50.7 | 61.8 | 42.7 |
| KD | 69M | 25.2 | 99.3 | 51.0 | 27.4 | **59.8** | 36.2 | **39.8** | 50.7 | 58.5 | 42.8 |
| CoCoMix | 69M | **24.7** | **99.1** | **50.9** | **27.6** | 59.5 | **37.2** | 39.3 | **51.0** | **58.2** | **42.9** |
| NTP | 386M | 16.3 | 26.3 | 29.9 | 33.6 | **64.1** | 38.4 | 47.3 | 50.9 | 24.2 | 46.8 |
| KD | 386M | 16.4 | 24.6 | **29.1** | 33.6 | 63.6 | 38.0 | **48.5** | 51.4 | 23.4 | 47.0 |
| CoCoMix | 386M | **15.9** | **19.3** | **29.1** | **34.7** | 63.6 | **39.2** | 47.9 | **52.3** | **21.4** | **47.5** |
| NTP | 1.38B | 14.3 | 16.6 | 25.0 | 38.1 | 66.1 | 38.9 | 50.5 | 50.0 | 18.6 | 48.7 |
| KD | 1.38B | 14.2 | 16.6 | **24.9** | 37.4 | 66.7 | 39.0 | 50.1 | 52.3 | 18.5 | 49.1 |
| CoCoMix | 1.38B | **13.9** | **15.4** | **24.9** | **38.4** | **66.9** | **39.5** | **50.8** | **53.0** | **18.1** | **49.7** |

Our results also indicate that larger models (e.g., 386M and 1.38B) can benefit from using concepts extracted from a smaller 124M model, showing effective weak-to-strong supervision. Moreover, as shown in Figure 2, CoCoMix consistently improves the performance over the NTP on a billion-scale model. For instance, CoCoMix achieves similar performance to NTP while using 21.5% fewer tokens, demonstrating high sample efficiency. Finally, it is worth noting that the performance gain of using CoCoMix is increasing over the training steps, demonstrating strong generalization performance.

**Comparison with KD baseline.** We also compare CoCoMix with KD baseline across multiple scenarios, including (1) a stronger teacher model teaching a smaller student model; (2) weak-to-strong supervision, where a weaker teacher teaches a larger student model; and (3) distribution shift, where the student is trained on a corpus different from the teacher's pretraining distribution. As shown in Table 1, CoCoMix demonstrates improvements over KD in all considered model configurations. In particular, CoCoMix shows significant performance gain in the weak-to-strong supervision setup, e.g., improving average perplexity of 2.8 in 386M, while KD does not show great improvement. This arises because a weaker teacher can introduce noisy or suboptimal knowledge, especially when the student surpasses the teacher in capability (Rawat et al., 2024). This trend is also observable in Figure 4, where models trained with KD fall behind standard training midway through training as the student outpaces the teacher (especially in the distribution shift scenario). In contrast, CoCoMix selectively utilizes useful concepts, resulting in a consistent performance gain.

## 3.2 INTERPRETABILITY AND STEERABILITY OF COCOMIX

Another core advantage of CoCoMix is its interpretability and model steering. Specifically, as the model is trained to predict concepts in its hidden state, we can analyze which concepts it focuses on based on the concept predictions. Furthermore, by amplifying the magnitude of the predicted

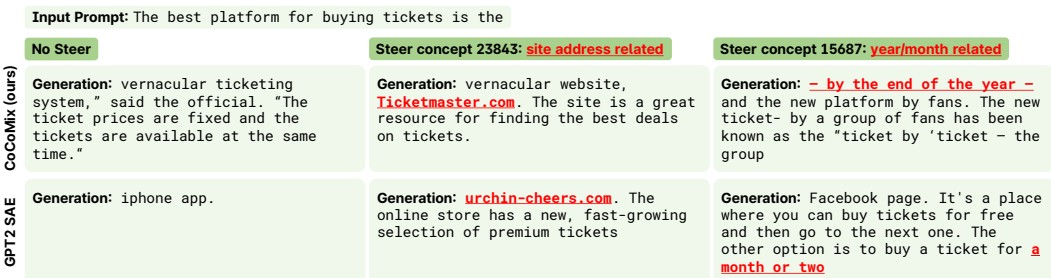

Figure 5: **Qualitative demonstration of the concept steering effect.** CoCoMix and GPT2 models are 350M and 124M parameter transformers, respectively, trained on the OpenWebText dataset. For CoCoMix, we manipulate the predicted concept logit $\mathbf{z}$, while for GPT2, we adjust the SAE concept space $\mathbf{c}$ by increasing the activation of a specific concept index. This illustrates the impact of targeted concept steering on the respective model outputs.

concept $\mathbf{z}_t$, one can control the output generation of the model. Following Templeton et al. (2024), we multiply $\mathbf{z}_t$ of a desired concept element by constants ranging from -10 to 10. To verify whether this steerability works as intended, we steer the activation of the same concept in the pretrained model's SAE latent space $\mathbf{c}$ and confirm whether the output exhibits the corresponding concept. Here, we use a 386M parameter model trained with CoCoMix, where the pretrained model is GPT-2. As shown in Figure 5, when the concept related to "website address" is amplified, both models start generating actual website addresses. This demonstrates that our model has successfully learned the GPT-2 aligned concepts. More examples of steering can be found in Appendix B.3.

## 3.3 ANALYSIS OF COCOMIX'S EFFECTIVENESS

In this section, we provide a detailed analysis of CoCoMix to validate the effect of each proposed component. Unless otherwise specified, we use a 69M model and train on 20B tokens.

**Effectiveness of the attribution score.** We first analyze whether the attribution score effectively extracts important concepts. To demonstrate this, we train CoCoMix using the activation value $\mathbf{c}_t$ for concept extraction, i.e., $\mathcal{L}_{\text{concept}}(\mathbf{c}_t)$ in Equation 2, instead of the attribution score $\mathbf{a}_t$. Remark that the activation value also well reflects the importance of the concept (Bricken et al., 2023). As shown in Figure 6a, using attribution scores significantly improves performance, improving sample efficiency by 17.5% compared to activation value based selection. We believe it will be an interesting future direction to explore other selection criteria for improving CoCoMix's performance or removing undesirable concepts to reduce bias, e.g., selectively removing unsafe concepts for safe training.

**Comparison with direct hidden state predictions.** To evaluate the importance of predicting the concept extracted from SAE, we compare CoCoMix with direct hidden state prediction strategies (i.e., predict the full hidden state without projecting into the concept space). To have a comparison under the same architecture as CoCoMix, we replace the concept prediction head $M$ with a two-layer multilayer perceptron (MLP), denoted $g(\cdot)$, which predicts the pretrained LLM's hidden state $\mathbf{h}^{\text{con}}$ directly from the hidden state of the model $\mathbf{h}$. The predicted representation, $g(\mathbf{h})$, is then compressed into a continuous embedding for insertion to have the same architecture as CoCoMix. Here, we use continuous loss including, $\ell_1$, $\ell_2$, and the cosine distance (e.g., $|\mathbf{h}^{\text{con}} - g(\mathbf{h})|_2^2$ for $\ell_2$) to predict the hidden state. As shown in Figure 6b, direct hidden state prediction leads to a performance drop. We conjecture this to be due to SAE's ability to decompose the latent into semantically meaningful concepts while predicting all hidden states may include noisy components, emphasizing the effectiveness of our method.

**Compression layer weight analysis.** Now, we analyze the weight of the compression layer $\mathbf{W}$ in Equation 3 to show how CoCoMix utilize the predicted concept. To this end, we visualize the $\ell_2$ norm of each concept's weights of 386M CoCoMix: for the weight matrix $\mathbf{W} \in \mathbb{R}^{d \times C}$, where $d$ is the hidden dimension and $C$ is the number of concepts, the norm is defined as: $\|\mathbf{W}_{:,c}\|_2 = \sqrt{\sum_{d=1}^{D} W_{d,c}^2}$. As shown in Figure 6c, we found that a portion of concept weights are near zero, e.g., 5.8% has a norm less than 1e-2. This indicates that CoCoMix ignores these concepts when compressing them into a

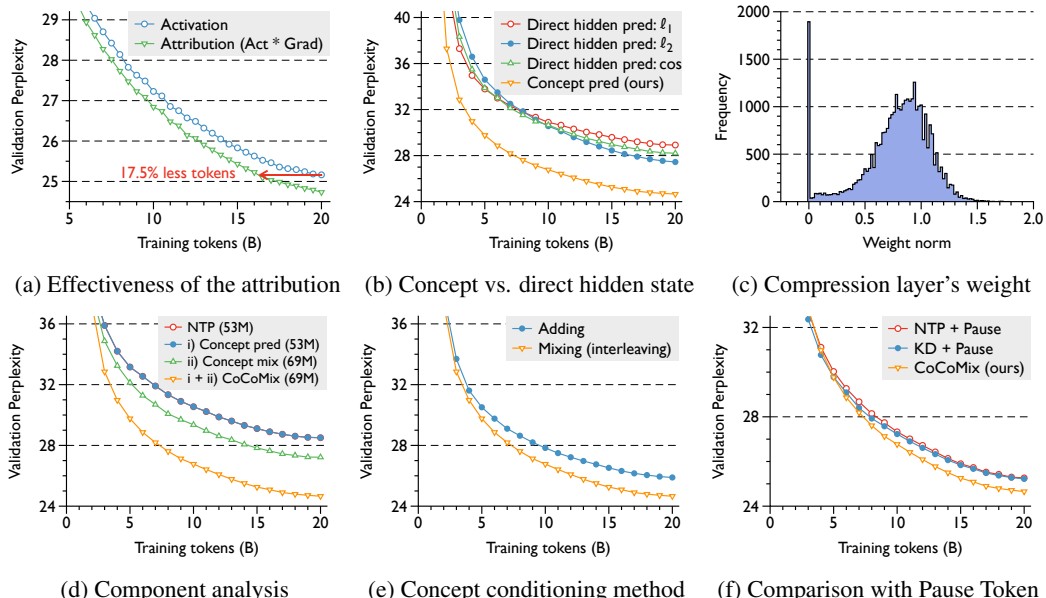

Figure 6: **Analysis of CoCoMix.** (a) Effectiveness of the attribution score for selecting concepts. (b) Comparison between concept prediction and direct hidden state prediction (i.e., predicting the hidden state with continuous loss rather than discretizing the hidden state with SAE). (c) The sparsity in the compression weight. (d) Component analysis by analyzing the contribution of concept prediction and mixing. (e) Design choices for concept conditioning by comparing adding the concept vector to the original hidden state and mixing (interleaving the concept vector with token hidden representation). (f) Comparison between CoCoMix and the Pause token (i.e., adding learnable tokens).

continuous concept if it is not useful. We conjecture this enabled CoCoMix for strong weak-to-strong supervision as it learned to ignore ineffective concepts extracted from a weak model.

**Component Analysis.** We analyze the contributions of each component of our method: (a) concept prediction Equation 2, and (b) concept insertion Equation 4. The results in Figure 6d demonstrate that both components are critical for performance improvement. Specifically, applying the concept prediction loss alone yields a modest reduction in perplexity. However, incorporating concept insertion alongside prediction enhances the effectiveness of the loss, resulting in further performance gains. This highlights the role of insertion in enabling the model to leverage the pretrained LLM's latent reasoning effectively. Notably, while concept insertion increases parameter count, it has a limited impact on performance when used alone, emphasizing the critical role of concept prediction.

**Computational cost analysis.** One possible concern of CoCoMix is the increased computation due to the expanded token sequence in upper layers. To this end, we report the total FLOPs required for training CoCoMix and baselines, including the NTP and the Pause token. As shown in Table 2, CoCoMix introduces a modest FLOPs increase compared to NTP, but remains more efficient than the Pause token while achieving better perplexity in both low-scale and large-scale settings. Furthermore, it is worth noting that CoCoMix with 140.7B tokens matches the perplexity of NTP trained with 200B tokens, using 17.8% fewer FLOPs, highlighting its improved efficiency–performance trade-off.

Table 2: **Computational cost.** Training FLOPs ($\times 10^{18}$) and validation perplexity (Val. PPL) comparison of NTP, Pause token, and CoCoMix, using a 69M model.

| Method | Train Tokens | FLOPs | Val. PPL |
|---|---|---|---|
| NTP | 20 B | 2.88 | 25.3 |
| + Pause token | 20 B | 3.48 | 25.1 |
| CoCoMix | 20 B | 3.37 | **24.7** |
| NTP | 200 B | 28.83 | 22.7 |
| CoCoMix | 141 B | **23.69** | 22.7 |
| CoCoMix | 200 B | 33.68 | **22.0** |

**Continuous concept mixing method.** We explored two methods for introducing the continuous concept $\hat{c}_t$ into the model's hidden state $h_t$, referred to as concept conditioning. The first method, insertion, insert the continuous concept in front of the token embedding, thus enabling the model to have a mixed input of the token and concept. The other option is intervention, which directly modifies the hidden state by adding the concept vector, i.e., $h_t \leftarrow h_t + \hat{c}_t$. As illustrated in Figure 6e,

both methods enhance pretraining performance (compared to NTP in Figure 6d), highlighting the importance of concept conditioning, where the insertion method performed better. By introducing a distinct concept vector, the insertion method enables the model to explicitly recognize and effectively utilize concepts during generation, enhancing its overall performance.

**CoCoMix vs. pause tokens.** Furthermore, we consider an additional baseline that jointly uses pause token (Goyal et al., 2024). Specifically, the pause token uses an additional learnable token that is inserted into the token embedding, enabling the LLM to think more before predicting the next token, which is similar to our continuous concept insertion. To this end, we insert the pause token for every input token on the same hidden state layer as CoCoMix to ensure comparable computation. Moreover, we also consider training the pause token with KD. As shown in Figure 6f, CoCoMix consistently outperforms the pause token, indicating our inserted (or interleaved) continuous concept indeed consists of useful information to improve the performance.

## 4 RELATED WORK

**Beyond token-level guidance for language modeling.** While next token prediction (NTP) remains the standard paradigm, recent works explore guidance beyond language tokens. For instance, some methods explore a better target, such as leveraging multi-token predictions to capture long context dependencies (Gloeckle et al., 2024; DeepSeek-AI, 2024) or predicting sequence embedding (Lee et al., 2024). Additionally, methods explore new types of inputs, e.g., using latents (Hao et al., 2024) or self-generated thought as inputs (Zelikman et al., 2024), which have shown improving reasoning capabilities. Only recently, concept-level modeling using local encoder-decoder architectures has also been explored to represent a language at a higher abstraction level (LCM et al., 2024). Other methods add extra tokens in the input space to increase computation at inference time Nye et al. (2021); Wei et al. (2022); Goyal et al. (2024); Lanchantin et al. (2024). In contrast, we propose a pretraining approach that integrates NTP with continuous concepts, connecting high-level semantics with fine-grained token guidance.

**Sparse Autoencoders (SAEs).** SAEs extend the autoencoder by enforcing sparsity constraints in the latent space (Lee et al., 2006). The features learned by SAEs are often interpretable and disentangled, making them useful across various domains, including language modeling (Bricken et al., 2023). Additionally, SAEs have gained attention in mechanistic interpretability due to their ability to capture coherent semantic concepts (Marks et al., 2024). This property has enabled practical advancements in identifying and manipulating semantic concepts and facilitating steering for controlled model outputs (Lieberum et al., 2024). Among SAE variants, TopK SAEs (Makhzani and Frey, 2014) enforce explicit sparsity using a TopK activation function, demonstrating effectiveness even for large models (Gao et al., 2024). In this work, we leverage SAE and, to the best of our knowledge, are the first to apply it to LLM pretraining, achieving strong performance while enhancing the interpretability and controllability of the trained model.

**Knowledge distillation (KD).** Our method can also be related to KD, i.e., transfers the expertise of a teacher model to a student model to enhance performance (Hinton et al., 2015; Zagoruyko and Komodakis, 2017), as CoCoMix extracts high-level semantic features from a pretrained model. Recently, KD for LLMs has garnered increasing attention, leveraging knowledge from a teacher to improve the generative and encoding capabilities of a student (Sanh et al., 2019; Ko et al., 2024). Especially, applying KD to LLM pretraining remains challenging due to the massive token scales (billions to trillions), forcing most current methods to resort to naive token-level probability matching (Team, 2024; Gu et al., 2024). Additionally, while pretrained models contain a vast amount of learned information and are thus beneficial to use, reusing knowledge from smaller teacher models remains challenging (Burns et al., 2023). In this work, we show CoCoMix can even leverage the concept extracted from small models to train a large model showing weak-to-strong supervision.

## 5 CONCLUSION

We propose CoCoMix, a new LLM pretraining framework that augments next token prediction with continuous concepts. By leveraging concepts extracted from a pretrained SAE as targets, our model predicts both the next token and the associated concept. Predicted concepts are then compressed into a continuous concept vector, which is then mixed into the hidden state. This approach enhances

interpretability and controllability by enabling direct probing of the distilled concepts. Experimental results show that CoCoMix consistently improves performance across benchmarks, particularly in challenging generalization scenarios such as weak-to-strong supervision.

## ETHICS STATEMENT

We introduce a novel LLM pretraining framework that augments continuous concepts for the next token prediction. This framework not only improves the performance but also enhances the interpretability and controllability of the LLM. By enabling direct probing and analyzing the CoCoMix's concept space, one can achieve transparency in understanding and guiding model behavior. Additionally, the framework offers the potential to selectively exclude harmful or undesirable concepts during LLM pretraining, contributing to the development of more responsible and ethical AI systems. We believe this research provides new perspectives on language modeling through enhanced interpretability.

## REPRODUCIBILITY STATEMENT

All implementation details, including the full set of hyperparameters and experimental settings, are described in Section 3 and Appendix A.

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

## A   EXPERIMENTAL DETAILS

**Architecture details.** Our method and baseline both utilize the GPT-2-based Transformer architecture and tokenizer (Radford et al., 2019), with a context length of 1024. We consider three model sizes, defined by the number of activated parameters: 69M, 386M, and 1.38B. For CoCoMix, the hidden state dimensions $d$ are 512, 1024, and 2028, with 8, 24, and 24 layers, respectively. The number of attention heads is set to 8, 16, and 16 for the three configurations. Baseline models are configured to match the number of activated parameters by employing the same number of layers and attention heads, with hidden state dimensions of 624, 1072, and 2096, respectively. We leverage an open-source pretrained TopK Sparse Autoencoder (SAE) (Gao et al., 2024) for concept extraction, where $K_{\texttt{concept}}$ is set to 32 and the concept activation size is 32,768. Consequently, our models introduce an additional $(C + K_{\texttt{concept}}) \times d$ activated parameters on top of the base Transformer parameters. The GPT-2 model (a teacher model for KD and a concept extraction model for CoCoMix) has 12 layers, a hidden dimension size of 768, and 124M parameters. The concept extraction layer $L_{\texttt{con}}$ is configured as the 6th middle layer of GPT-2 for all model sizes. For CoCoMix, the concept prediction is done at the 4th layer for the 69M model and the 6th layer for the larger 386M and 1.38B models. The reported training FLOPs in Table 2 are based on the base model, and show low FLOPs thanks to the high sparsity of the compression layer (only activated with a concept number of 32), which introduces a small cost.

**Training details.** We mainly followed the training details outlined in (Brown et al., 2020). For the main results presented in Section 2, models were trained on 200B tokens over approximately 382K training steps, while other experiments were conducted on 20B tokens. We used OpenWebText as the training dataset to match the same training corpus as GPT-2. The learning rate schedule included a warm-up phase over the first 1/300 of the total training steps, followed by a cosine decay to 10% of the maximum learning rate at the end of training. The maximum learning rates were set to 6e-4, 3e-4, and 2e-4 for the 69M, 386M, and 1.38B models, respectively. The batch sizes were configured to 0.5M, 0.5M, and 1M tokens for the three model sizes. A weight decay of 0.1 was applied across all configurations, and we utilized the AdamW optimizer (Loshchilov, 2017) with $\beta_1 = 0.9$ and $\beta_2 = 0.95$. For training CoCoMix, the concept prediction loss Equation 4 was scaled by a hyperparameter $\lambda = 0.1$, and $K_{\texttt{attri}}$ was set to 4. Here, as the next token prediction (NTP) loss is our primary objective, we chose $\lambda$ that is smaller than 1 to ensure that the concept prediction loss—which has a similar scale—would not dominate but instead serve as an auxiliary objective. We found that this value empirically worked well with our small-scale model (69M), and thus we consistently used 0.1 throughout all experiments. While further tuning could potentially yield better results, this initial choice was sufficient to clearly demonstrate our method's effectiveness. For training CoCoMix under a 69M transformer for 20B tokens, we have used 16 A100 GPUs for less than 8 hours.

For the KD baseline, we employed the vanilla KD loss, where the output probabilities of the teacher model $\mathcal{M}_{\texttt{con}}$ and the student model $\mathcal{M}$ were matched using the Kullback-Leibler (KL) divergence. Specifically, given an input $\mathbf{x}$, the KD loss is defined as $-\log \mathcal{M}(x_{t+1}|\mathbf{x}) + \lambda_{\text{KD}} \cdot \text{KL}(\mathcal{M}_{\texttt{con}}(\cdot|\mathbf{x}) \| \mathcal{M}(\cdot|\mathbf{x}))$, where $\lambda_{\text{KD}} = 0.1$ consistently demonstrated strong performance.

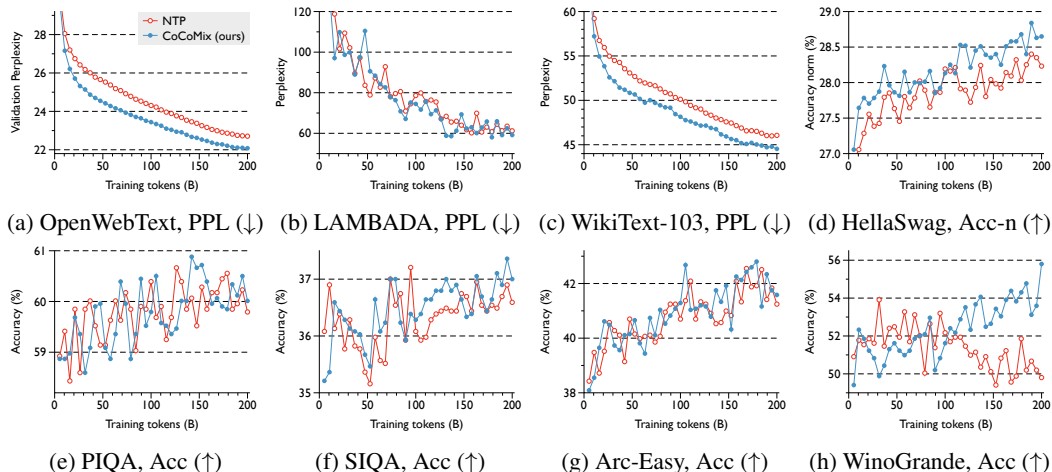

(a) OpenWebText, PPL (↓)   (b) LAMBADA, PPL (↓)   (c) WikiText-103, PPL (↓)   (d) HellaSwag, Acc-n (↑)

(e) PIQA, Acc (↑)   (f) SIQA, Acc (↑)   (g) Arc-Easy, Acc (↑)   (h) WinoGrande, Acc (↑)

Figure 7: **CoCoMix vs. NTP performance at different training checkpoints on 69M parameter model.** Each model is trained on the 200B tokens sampled from the OpenWebText dataset. The plot shows the result of (a) OpenWebText, (b) LAMBADA, (c) WikiText-103, (d) HellaSwag, (e) PIQA, (f) SIQA, (g) Arc-Easy, and (h) WinoGrande datasets. We use the concepts extracted from a 124M-sized model for training CoCoMix.

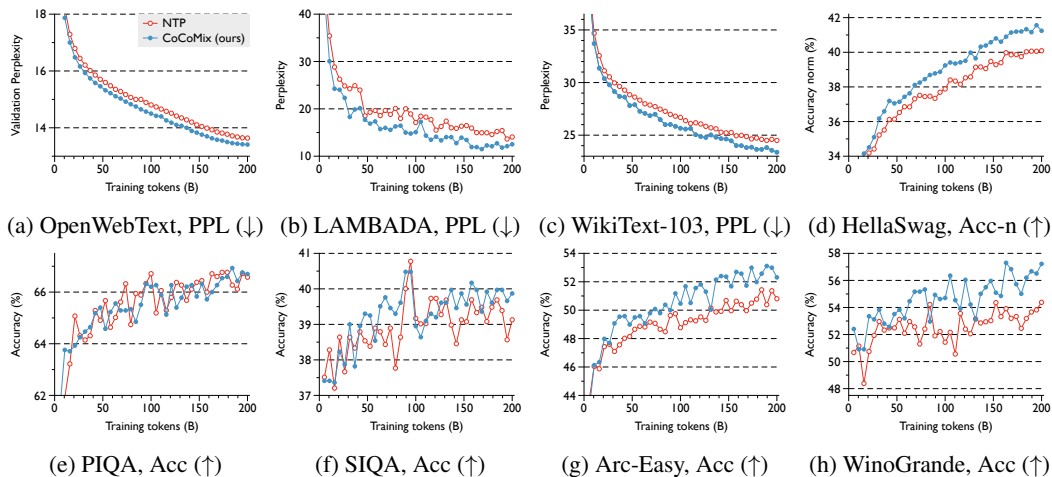

(a) OpenWebText, PPL (↓)   (b) LAMBADA, PPL (↓)   (c) WikiText-103, PPL (↓)   (d) HellaSwag, Acc-n (↑)

(e) PIQA, Acc (↑)   (f) SIQA, Acc (↑)   (g) Arc-Easy, Acc (↑)   (h) WinoGrande, Acc (↑)

Figure 8: **CoCoMix vs. NTP performance at different training checkpoints on 368M parameter model.** Each model is trained on the 200B tokens sampled from the OpenWebText dataset. The plot shows the result of (a) OpenWebText, (b) LAMBADA, (c) WikiText-103, (d) HellaSwag, (e) PIQA, (f) SIQA, (g) Arc-Easy, and (h) WinoGrande datasets. We use the concepts extracted from a 124M-sized model for training CoCoMix.

## B   ADDITIONAL RESULTS

### B.1   DETAILED PERFORMANCE DURING TRAINING

In this section, we present the performance tracking during training on 200B tokens, including validation perplexity and the perplexity and accuracy of various downstream tasks, including LAMBADA, WikiText-103, HellaSwag, PIQA, SIQA, Arc-Easy, and WinoGrande datasets. As shown in Figure 7, Figure 8, and Figure 9, we compare CoCoMix with the next token prediction (NTP) across different active parameter sizes: 69M, 386M, and 1.38B. In most of the graphs, CoCoMix consistently demonstrates performance gains. Notably, our results show that CoCoMix achieves stable improvements in perplexity across all tasks. Furthermore, CoCoMix exhibits sample efficiency;

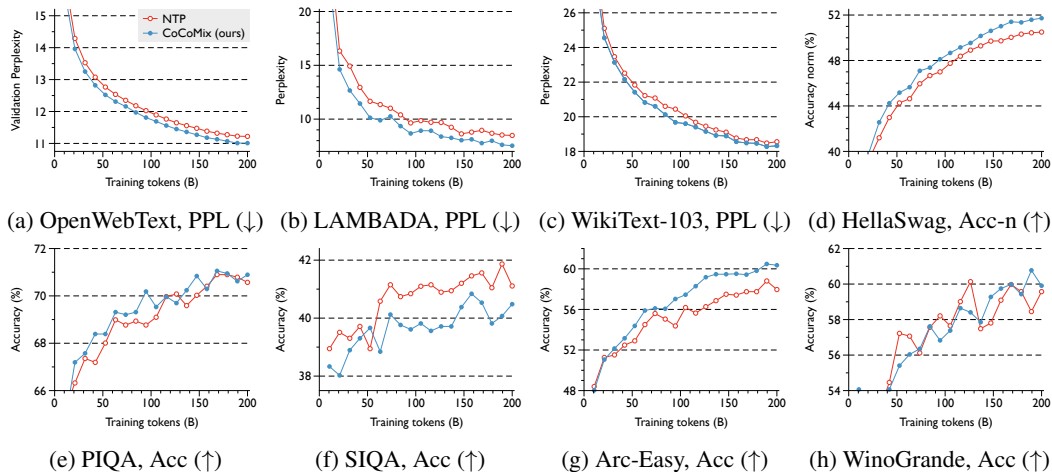

Figure 9: **CoCoMix vs. NTP performance at different training checkpoints on 1.38B parameter model.** Each model is trained on the 200B tokens sampled from the OpenWebText dataset. The plot shows the result of (a) OpenWebText, (b) LAMBADA, (c) WikiText-103, (d) HellaSwag, (e) PIQA, (f) SIQA, (g) Arc-Easy, and (h) WinoGrande datasets. We use the concepts extracted from a 124M-sized model for training CoCoMix.

Table 3: **Comparison of CoCoMix with ablation baselines.** We report validation perplexity on OpenWebText using a 69M-parameter Transformer model.

| Method | Validation PPL |
|---|---|
| NTP (token-only) | 25.3 |
| (i) Interleave same token twice | 25.1 |
| (ii) Insert concept every $T = 4$ steps | 25.0 |
| (iii) Concept-only (no token hidden states) | 25.1 |
| **CoCoMix (Ours)** | **24.7** |

for instance, in the 1.38B model, CoCoMix reaches the same OpenWebText validation perplexity as NTP while requiring approximately 43B fewer tokens (a 21.5% improvement in token efficiency).

## B.2 MORE COMPARISON WITH ADDITIONAL BASELINES

To further demonstrate the effectiveness of CoCoMix, we consider three additional baselines. Specifically, we compare with: (i) duplicating each token hidden state, resulting in an interleaved sequence of $(\mathbf{h}_1, \mathbf{h}_1, \mathbf{h}_2, \mathbf{h}_2, \cdots, \mathbf{h}_t, \mathbf{h}_t)$; (ii) interleaving continuous concepts every $T$ steps instead of at every time step, yielding $(\mathbf{h}_1, \cdots, \mathbf{h}_T, \hat{\mathbf{c}}_T, \mathbf{h}_{T+1}, \cdots, \mathbf{h}_{2T}, \hat{\mathbf{c}}_{2T}, \cdots)$; where we use $T = 4$ and (iii) using only continuous concepts without token hidden states, i.e., $(\hat{\mathbf{c}}_1, \hat{\mathbf{c}}_2, \cdots, \hat{\mathbf{c}}_t)$.

This comparison isolates key components of CoCoMix and highlights their contributions to overall performance. We present the result in Table 3. First, duplicating token hidden states to match the sequence length of CoCoMix yields worse performance despite higher computational cost (it is worth noting that the copied baseline increases the forward GFLOPs even higher than CoCoMix, 164.46 → 169.52 for computing 1024 tokens), indicating that gains are not solely due to sequence length. Second, reducing the interleaving frequency of concept vectors leads to performance degradation, suggesting that frequent integration of semantic signals is crucial. Lastly, a model using only continuous concepts outperforms token-only models but still underperforms CoCoMix, confirming that the synergy between token-level and concept-level representations is essential for optimal effectiveness.

## B.3    Additional Steerability Results

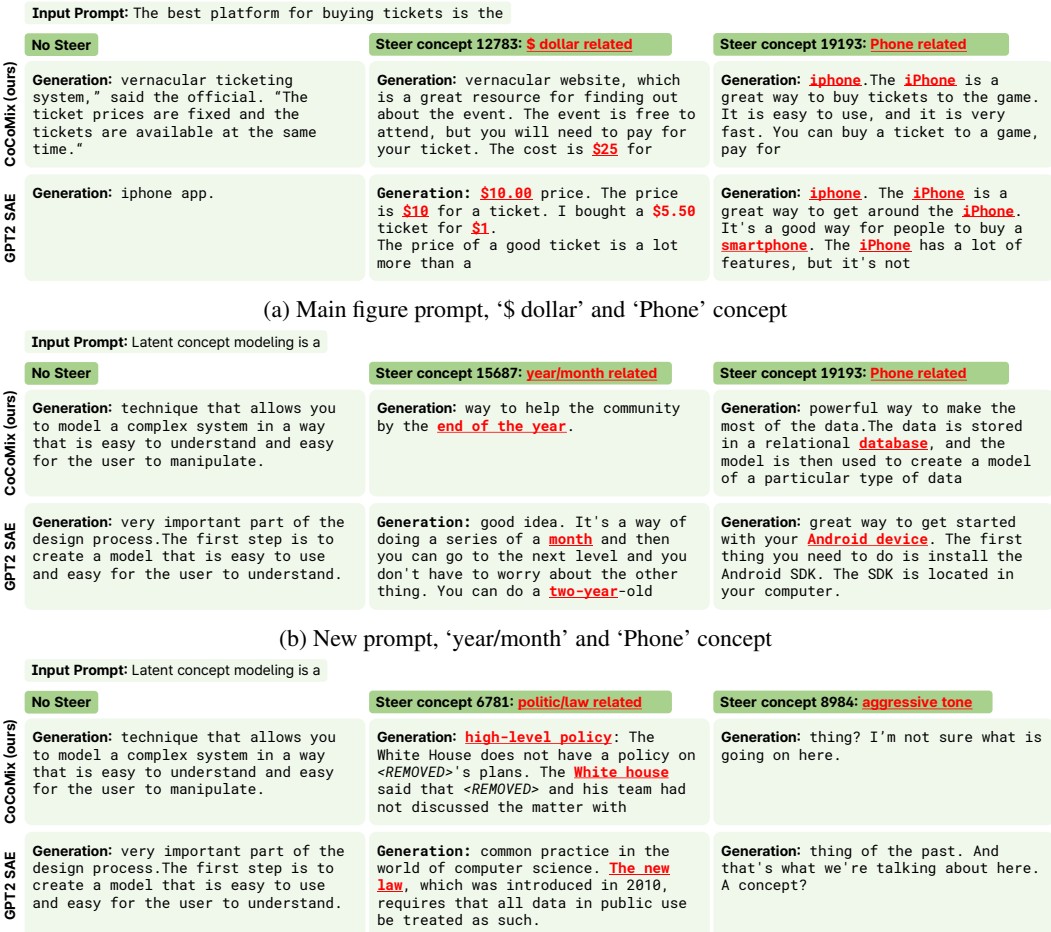

(a) Main figure prompt, '$ dollar' and 'Phone' concept

(b) New prompt, 'year/month' and 'Phone' concept

(c) New prompt, 'Politic/law' and 'aggressive tone' concept

Figure 10: More qualitative demonstration of the concept steering effect. CoCoMix and GPT2 models are 350M and 124M parameter transformers, respectively. For CoCoMix, we manipulate the predicted logit **z**, while for GPT2, we adjust the SAE concept space **c** by increasing the activation of a specific concept index

To further analyze the steerability enabled by CoCoMix, we conducted experiments using both the same prompt as in the main figure and a new prompt (in Figure 10). For consistency, we first applied steering on additional concepts identified during our analysis—"$ dollar" and "Phone"—using the same prompt as in Figure 5. These experiments confirmed that the model could effectively modulate its output based on these newly identified concepts, producing coherent and concept-aligned generations. Next, to verify whether the identified concepts generalize to different contexts, we experimented with a new prompt: *"Latent concept modeling is a"* and steered the model using the previously identified concepts "month/year" and "Phone." The results showed that the model successfully reproduced outputs aligned with these concepts, further supporting the robustness of our method. Additionally, we explored whether new concepts could be identified and steered using the same prompt. In this case, we identified two new concepts: "politics/law" and "aggressive tone." Steering the model with these new concepts demonstrated that the outputs could be effectively controlled to exhibit characteristics aligned with the corresponding concepts. These findings further highlight the flexibility and interpretability of our approach.

## C FUTURE WORK AND LIMITATIONS

Several recent approaches, including ours, leverage additional compute to support more advanced forms of reasoning, such as test-time optimization or latent thinking via pause tokens. While this often leads to improved performance, it naturally increases computational cost. As shown in Section 3.3, CoCoMix achieves favorable compute efficiency, demonstrating stronger performance per FLOP compared to prior latent thinking methods. This opens up an interesting direction for future work: designing more informative and structured inputs that better utilize the compute budget of the Transformer to improve the performance. Also, future work could explore learning continuous concepts during pretraining without the need for distillation.

