# OpenReview forum: "LLM Pretraining with Continuous Concepts"
_ICLR.cc/2026/Conference — ICLR 2026 Poster_

### Official Review · Reviewer_LpV1 · 2025-10-29

**Soundness:** 3
**Presentation:** 3
**Contribution:** 3
**Rating:** 8
**Confidence:** 3

**Summary:**

In this paper, the authors design a new method to improve the next-token prediction and final performance. Specifically, they design CoCoMix, a new pretraining framework for large language models that augments standard next-token prediction with an additional objective: predicting “continuous concepts” derived from a sparse autoencoder and injecting those concepts directly into the model’s hidden states, interleaved with normal token representations. It improves both sample efficiency and downstream performance on language modeling and reasoning tasks. Beyond accuracy, CoCoMix also makes models more interpretable and steerable, because the predicted concepts can be inspected and edited to transparently influence the model’s internal reasoning. The provided figures and examples make this paper easier to understand. Overall, the quality of this paper will be further improved after addressing concerns listed below.

**Strengths:**

The topic is highly interesting and might generate broad impact to the LLM community.

I like the visualizations of figures, which are clear and improve the readability of this paper. The reviewer appreciates the authors for doing this.

In experiments, the analysis is solid and comprehensive.

**Weaknesses:**

[1 ] To be honest, after reading Figure 1 alone or combined with the text in the introduction, the reviewer is still confused about how the extracted concepts benefit the next token prediction. More explanations might be helpful.

[2] During the target concept selection process, it the attribution conducted in each training batch?

[3] Another question is about clarification, after the concept selection, why it is necessary to conduct concept prediction?

[4] In experiments, the model sizes are relatively small. The reviewer understand this might be because the limited budget in model training. Could you discuss whether the findings in this paper could generalize to larger models?

[5] It will be great to disclose the training costs, e.g., GPU numbers, types, GPU hours.

[6] While there are performance gain, will it brought other costs like training time when applying COOMIX?

[7] Sumamrizing the performance gain will improve this paper further. For example, in the abstract, intro, captions of Figure 2 and 3, it will be great to introduce the performance gain.

[8] The steering example is interesting. COuld you introduce how do we know which concept is related to “website” or “money”? The transparency will improve the readability of this paper.

**Questions:**

[1 ] To be honest, after reading Figure 1 alone or combined with the text in the introduction, the reviewer is still confused about how the extracted concepts benefit the next token prediction. More explanations might be helpful.

[2] During the target concept selection process, it the attribution conducted in each training batch?

[3] Another question is about clarification, after the concept selection, why it is necessary to conduct concept prediction?

[4] In experiments, the model sizes are relatively small. The reviewer understand this might be because the limited budget in model training. Could you discuss whether the findings in this paper could generalize to larger models?

[5] It will be great to disclose the training costs, e.g., GPU numbers, types, GPU hours.

[6] While there are performance gain, will it brought other costs like training time when applying COOMIX?

[7] Sumamrizing the performance gain will improve this paper further. For example, in the abstract, intro, captions of Figure 2 and 3, it will be great to introduce the performance gain.

[8] The steering example is interesting. COuld you introduce how do we know which concept is related to “website” or “money”? The transparency will improve the readability of this paper.

---

> ### Author Response · Authors · 2025-11-21
>
> Dear reviewer LpV1,
>
> We sincerely appreciate your efforts and insightful comments to improve the manuscript. We respond to each of your comments one-by-one in what follows.
>
> ---
>
> **[Q1] How does the concept benefit the next token prediction?**
>
> In CoCoMix, we show that SAE concepts are not only useful for analyzing LLM behavior but also become explicit supervision and conditioning signals: we first use attribution to select the concepts whose change most affects the next token log probability, train the model to predict these selected concepts from its hidden state, and then compress and interleave the predicted concept vector with the token representations so that the decoder can directly use this high level summary when producing the next token. Intuitively, the model is encouraged to first infer a compact set of semantic factors that are causally important for the next token and then use them as an additional latent thought that guides token prediction, which leads to better perplexity and data efficiency as shown in Figures 2–4 and the ablations in Section 4.3.
>
> ---
>
> **[Q2] During the target concept selection process, is the attribution calculation performed at every training batch?**
>
> This is an implementation design choice. In our current implementation, we compute the attribution batch-wise for simplicity of implementation. However, to make the process more efficient and scalable, one can simply pre-extract the selected concepts from the SAE and store them, allowing training to only load the cached concepts. This precomputation and lookup strategy is also commonly adopted in recent large-scale distillation pipelines (e.g., Gemma 2 [1]).
>
> [1] Gemma 2: Improving Open Language Models at a Practical Size
>
> ---
>
> **[Q3] Why is it necessary to conduct concept prediction?**
>
> Conducting concept prediction is necessary because it ensures the model internalizes which SAE concepts are causally important for the next token. By predicting the top-attribution concepts from its own hidden state, the model aligns its representation with the concept space and learns a compact high-level summary that can then be compressed and interleaved, which empirically leads to better perplexity and data efficiency. Our ablations (Figure 6b) further show that removing the concept-prediction loss largely eliminates the gains of CoCoMix, indicating that this step is central to the method’s effectiveness.
>
> ---
>
> **[Q4] Could the findings generalize to larger models, given that the experiments use relatively small models due to budget constraints?**
>
> While we agree that demonstrating results at a larger scale would have been desirable, we operated under practical budget constraints and therefore presented results up to the billion parameter range, which was the largest feasible setting for us. We note that one billion parameter scale models are a widely accepted academic standard [2,3,4], and many prior studies, including those that ultimately train much larger models such as Llama 3 [5], analyze scaling trends from millions to billions as a critical indicator of method behavior. We therefore believe that CoCoMix is likely to follow a similar trajectory, and we look forward to exploring its behavior at even larger scales in future work.
>
> [2] Physics of language models: Part 4.1, architecture design and the magic of canon layers\
> [3] Block Transformer: Global-to-Local Language Modeling for Fast Inference, NeurIPS 2024\
> [4] Think before you speak: Training Language Models With Pause Tokens, ICLR 2024\
> [5] The Llama3 Herd of Models
>
> ---
>
> **[Q5] It will be great to disclose the training costs, e.g., GPU numbers, types, GPU hours.**
>
> To give a sense of scale, our compute usage increases smoothly with model size: the 69M model was trained on 20B tokens using 16×A100 (80GB) GPUs for under 8 hours, and the 1.3B model required 32×A100 GPUs for about 42 hours on the same dataset. Extending this setup to the full 200B-token run results in roughly 10× more total compute. This slowdown partly reflects our batch-wise concept extraction, and as noted in [Q2], a pre-compute-and-load approach would reduce the overall training time.
>
> ---

---

> ### Author Response · Authors · 2025-11-21
>
> ---
>
> **[Q6] While there are performance gains, will it bring other costs like training time when applying COOMIX?**
>
> While CoCoMix does introduce extra computation beyond plain next token prediction, the key point is that this overhead is modest and largely decoupled from the main training loop. In Table 2, we report the training FLOPs of the student model only, which already include the concept prediction head and the interleaving operations, and CoCoMix still achieves better perplexity than NTP at matched student FLOPs. In addition, CoCoMix requires a separate offline pass to extract SAE concept labels using a fixed 124M GPT2 SAE teacher. Following common practice in knowledge distillation, we reuse a public pretrained teacher and do not count its pretraining cost; the extraction pass itself uses a much smaller model, can be fully precomputed and cached once for a given corpus, and reused across CoCoMix runs. It is worth noting that as data becomes increasingly scarce for pretraining, the improved data efficiency brought by CoCoMix provides a favorable tradeoff relative to this modest additional cost.
>
> ---
>
> **[Q7] Summarizing the performance gain in the Figure caption will improve the paper.**
>
> We thank the reviewer for the helpful suggestion. In the revised manuscript, we now include the performance gains directly in the figure caption for clarity, with the newly added text marked in blue.
>
> ---
>
> **[Q8] How can one identify which concept is related to “site” or “money”?**
>
> We thank the author for the question. According to the pretrained SAE feature codebook  [6,7], concept 23843 has the highest affinity for tokens related to site addresses, which is why it manifests as a “site‑address” concept in the visualization. For your interest, some works on mechanistical interpretability use LLMs to automatically identify such concepts [8].
>
> [6] Scaling and evaluating sparse autoencoders, ICLR 2025\
> [7] https://openaipublic.blob.core.windows.net/sparse-autoencoder/sae-viewer/index.html\
> [8] Automatically Interpreting Millions of Features in Large Language Models, ICML 2025
>
> ---

---

> > ### Comment · Reviewer_LpV1 · 2025-11-27
> >
> > THank you for the detailed responses, which have addressed most of my concerns. While I still think the impact of this paper will be further enhanced by evaluating on larger models, I do understand the limited budget constraint. Therefore, I still support the acceptance of this paper. Looking forward to seeing the camera-ready version.

---

### Official Review · Reviewer_XEZ8 · 2025-10-30

**Soundness:** 3
**Presentation:** 3
**Contribution:** 3
**Rating:** 6
**Confidence:** 3

**Summary:**

This work includes a range of experiments that demonstrate the model’s performance across multiple dimensions, such as downstream accuracy, efficiency, and steerability. Overall, the experiments are clearly presented and thoughtfully discussed. The chosen baselines are relevant, though somewhat limited. The analysis of the CocoMix model’s effectiveness is generally solid and supported by clear results.

**Strengths:**

1. The CoCoMix model is well described, and the method is easy to follow.

2. The proposal of combining next token prediction with continuous concepts in the pretraining paradigm is novel. This idea of integrating an interpretability mechanism (concept) into pretraining frameworks through SAE comes with significant originality. Such pretraining innovations remain rare in the field, and the model's effectiveness suggests potential incremental impact.

3. This work includes a range of experiments that demonstrate the model’s performance across multiple dimensions, such as downstream accuracy, efficiency, and steerability. Overall, the experiments are clearly presented, and the chosen baselines are relevant, though somewhat limited. The analysis of the CocoMix model’s effectiveness is generally solid.

**Weaknesses:**

1. Concept Interpretability: This work has been based on the assumption that the latent representation layer in SAE corresponds to human-interpretable concepts. Since this is central to the interpretability claims, more content addressing this assumption would be helpful, beyond what is described in the steerability section. How exactly does CoCoMix capture real, continuous mixture of concepts?

2. Model Design Justifications: Some architectural choices could be better justified if the authors add more explanations or ablation studies. For example, what's the significance of compressing top concepts into a "continuous" one, instead of focusing on the top concept?

3. The authors claim that their improvement on downstream tasks shows that CocoMix is better than NTP (Figure 3), but these differences in performance are minimal at best, and I'm not sure if they're significant. Is it worth the overhead?

**Questions:**

1. Figure 2 PPL scores -- are they computed using NTP objective for both models, or using NTP for the NTP baseline and the CocoMix PPL for the proposed method? If it's the latter, these scores are not really comparable...
2. For reproducibility reasons, I suggest the authors provide a link to their repo containing the code for the implemented methods and the experiments.
3. Does CoCoMix scale to larger LLMs?
4. Are the differences reported in the paper statistically significant?

---

> ### Author Response · Authors · 2025-11-21
>
> Dear reviewer XEZ8,
>
> We sincerely appreciate your efforts and insightful comments to improve the manuscript. We respond to each of your comments one-by-one in what follows.
>
> ---
>
> **[W1] Clarification on the assumption that SAE latents are human-interpretable and on how CoCoMix captures such concepts.**
>
> We would like to emphasize that the assumption that SAE latent features are human-interpretable is not a claim introduced by our work; but rather an established premise repeatedly supported in prior SAE studies [1,2,3,4,5]. Building on this, our contribution is to show that these concepts are not only human-interpretable, but also provide useful supervision signals beyond text-token prediction, leading to improved downstream performance. Furthermore, we provide empirical evidence that CoCoMix can effectively capture such concepts by predicting the same underlying SAE features: for example, the CoCoMix dimension associated with the “site’’ feature in SAE steers generation toward “site’’-related text when its activation is increased, consistent with the intended meaning of that concept (in Figure 5).
>
> [1] Scaling and evaluating sparse autoencoders, ICLR 2025\
> [2] Scaling Monosemanticity: Extracting Interpretable Features from Claude 3 Sonnet\
> [3] Gemma scope: Open sparse autoencoders everywhere all at once on gemma 2\
> [4] Automatically Interpreting Millions of Features in Large Language Models, ICML 2025\
> [5] Towards Monosemanticity: Decomposing Language Models With Dictionary Learning
>
> ---
>
> **[W2] Clarification on CoCoMix design choices, including the continuous concept representation.**
>
> We have discussed multiple design choices of CoCoMix, including the ones mentioned by the reviewer, in Figure 6 and Appendix B.2. Although we tested directly inserting the top K concepts, this approach increased inference FLOPS and consistently underperformed the compressed continuous representation (in Table 3). We believe this is because inserting the top K concepts implicitly requires the model to preserve the ordering and relative importance of these concepts during inference, whereas compression provides a more stable inductive bias without increasing the number of tokens. Our ablation studies, such as insertion versus addition (in Figure 6) and using only the continuous concept without token-level hidden states (in Table 3), highlight the different design components, and support the choices used in the final CoCoMix runs.
>
> ---
>
> **[W3] The improvement on downstream tasks seems modest.**
>
> Although the absolute downstream performance improvement is seemingly modest, we emphasize that CoCoMix achieves a 21.5 % gain in sample efficiency, i.e., a steeper perplexity reduction per training step. Note that the downstream accuracy heavily depends on the curation quality of the pre‑training corpus [6,7], where we deliberately used OpenWebText (i.e., a somewhat old dataset used to train GPT2) to follow well-constructed training setup for a fair comparison rather than newer datasets. We thus view the per‑step perplexity reduction as the more meaningful evidence, and attribute the remaining accuracy gap mainly to dataset quality rather than a limitation of our method. We demonstrated at a medium scale that our approach works reasonably well, and we believe that scaling it up can lead to more substantial gains.
>
> [6] The FineWeb Datasets: Decanting the Web for the Finest Text Data at Scale, 2024\
> [7] Data Selection for Language Models via Importance Resampling, NeurIPS 2023
>
> ---

---

> ### Author Response · Authors · 2025-11-21
>
> ---
>
> **[Q1] Clarification on how PPL is computed for fair comparison: Are the PPL scores in Figure 2 computed using the same NTP-based PPL for both models?**
>
> We confirm that all perplexity (PPL) numbers in Figure 2 (and in other results) are computed in exactly the same way across models. Specifically, we evaluate PPL using the standard next-token prediction (NTP) objective for both the NTP baseline and CoCoMix. Although CoCoMix is trained with an additional concept-prediction loss, this auxiliary objective is not used during evaluation; the likelihood is computed solely from the NTP head. Therefore, the reported PPL values are fully comparable across models and reflect the same metric.
>
> ---
>
> **[Q2] Implementation of the CoCoMix code.**
>
> We thank the reviewer for asking this question, and added an anonymous link as follows.
>
> https://anonymous.4open.science/r/cocomix_iclr2026-C6BD/README.md
>
> ---
>
> **[Q3] Does CoCoMix scale to larger LLMs?**
>
> Thank you for the thoughtful question. While demonstrating results at a larger scale would have been desirable, we operated under practical budget constraints and therefore presented models up to the billion-parameter range, which was the largest feasible setting. It is worth noting that 1B-scale models are a widely accepted academic standard [8,9,10], and numerous prior studies have shown that empirical findings at this scale tend to transfer reliably to larger models [11,12]. We believe that CoCoMix will follow this trajectory as well, and we look forward to examining its behavior at larger scales in future work.
>
> [8] Physics of language models: Part 4.1, architecture design and the magic of canon layers\
> [9] Block Transformer: Global-to-Local Language Modeling for Fast Inference, NeurIPS 2024\
> [10] Think before you speak: Training Language Models With Pause Tokens, ICLR 2024\
> [11] The Llama3 Herd of Models\
> [12] Training Compute-Optimal Large Language Models
>
> ---
>
> **[Q4] Are the differences reported in the paper statistically significant?**
>
> While we report experiments based on a single seed due to the significant computational cost associated with pretraining large models, the observed performance gains are consistent and reproducible across multiple model scales (e.g., 69M, 386M, and 1.38B). This consistency strongly suggests that our findings are robust rather than being artifacts of random initialization.
>
> ---

---

> > ### Comment · Reviewer_XEZ8 · 2025-11-26
> >
> > W1: This was not clear from the paper itself, thank you. I would advise the authors to clarify this in the paper. Moreover, while I appreciate the references to prior SAE work, my original concern was not whether SAEs can yield interpretable features, but whether the particular SAE features used here are sufficiently validated in the context of CoCoMix. The site example is helpful but limited in scope. The paper would benefit from a more systematic evaluation of which SAE features CoCoMix reliably captures, how often steering succeeds, and how robust this is across features.
> >
> > W2: The explanation about continuous compression versus top-K insertion is clearer now, but I still find the justification somewhat underdeveloped in the paper. Several design choices appear to be empirically motivated but are not conceptually motivated. I encourage the authors to expand the rationale, ideally adding a more principled discussion of why compression is expected to improve inductive bias beyond FLOP considerations.
> >
> > W3: Your abstract says "Through experiments on multiple benchmarks, including language modeling and downstream reasoning tasks, we show that CoCoMix is more sample efficient and consistently outperforms standard next token prediction and knowledge distillation." The rest of the paper follow this framing. If the authors wish to restate and stress the efficiency angle, that is acceptable. But please do not oversale your performance.
> >
> > Q3: The claim that results at 1B "tend to transfer" is not guaranteed, especially when introducing a new auxiliary loss. The paper would benefit from at least a discussion of potential scaling challenges (e.g., concept drift, SAE quality at scale).
> >
> > Q4: Compute constraints are understandable to some extend, but using only one seed leaves uncertainty about robustness. Consistency across model sizes does not substitute for variance estimates. Even small-scale, partial ablations with multiple seeds would provide stronger evidence. Moreover, some statistical tests would still work even for small data with one run.

---

### Official Review · Reviewer_SsAp · 2025-11-03

**Soundness:** 3
**Presentation:** 4
**Contribution:** 4
**Rating:** 8
**Confidence:** 4

**Summary:**

The authors propose a new pre-training paradigm based on "continuous concepts".

Specifically, the authors introduce an additional prediction head in a "standard" (transformer) LLM to produce continuous tokens based on "concepts". In order to train this additional prediction head the authors propose an additional training objective based on Sparse AutoEncoders (SAEs) and importance attribution using gradients.

The authors then validate their approach across various model sizes and baselines.

**Strengths:**

The main strenghts of the paper:
1. Great pre-training analysis with a novel architecture.
2. A working pre-training recipe that seems to improve performance
3. A way of introducing steerable concepts into the models generation. This can open the door to a lot of interesting research.

**Weaknesses:**

Main weaknesses:
1. Model sizes are limited (but understandable).
2. Hyper-parameter tuning was not discussed in detail and perhaps some of the scores can be attributed to poor hyper-params.

**Questions:**

Q1: How can you make sure your results are not attributed to randomness in hyper-param tuning? Can you demonstrate anything to this effect?

---

> ### Author Response · Authors · 2025-11-21
>
> Dear reviewer SsAp,
>
> We sincerely appreciate your efforts and insightful comments to improve the manuscript. We respond to each of your comments one-by-one in what follows.
>
> ---
>
> **[W1] Model sizes are limited (but understandable).**
>
> Thank you for the thoughtful question. While demonstrating results at a larger scale would have been desirable, we operated under practical budget constraints and therefore presented models up to the billion-parameter range, which was the largest feasible setting. It is worth noting that 1B-scale models are a widely accepted academic standard [1,2,3], and numerous prior studies have shown that empirical findings at this scale tend to transfer reliably to larger models [4,5]. We believe that CoCoMix may follow this trajectory as well, and we look forward to examining its behavior at larger scales in future work.
>
> [1] Physics of language models: Part 4.1, architecture design and the magic of canon layers\
> [2] Block Transformer: Global-to-Local Language Modeling for Fast Inference, NeurIPS 2024\
> [3] Think before you speak: Training Language Models With Pause Tokens, ICLR 2024\
> [4] The Llama3 Herd of Models\
> [5] Training Compute-Optimal Large Language Models
>
> ---
>
> **[W2 & Q1] Question about whether the results are sensitive to randomness in the hyperparameter selection process.**
>
> We carefully note that the only major hyperparameter in our framework is the weighting factor $\lambda$ for the concept‑prediction loss, which we selected using a simple heuristic rather than extensive tuning. Other hyperparameters were straightforward to determine; for example, we directly adopted Top‑K = 32 because the pretrained SAE checkpoint from prior work was built with that setting.
>
> In our limited sweep, we observed that training was stable for the entire range $\lambda\in [0, 1]$, indicating that the model is not particularly sensitive to $\lambda$. We fixed $\lambda=0.1$ because next‑token prediction (NTP) is our primary objective, and using a small $\lambda$ ensures that the concept‑prediction loss (which has a comparable raw scale) remains an auxiliary signal without overshadowing NTP. Moreover, $\lambda=0.1$ already provided strong gains on our 68 M model, so we used it consistently across all experiments.
>
> ---

---

### Meta-Review · Area_Chair_MnuR · 2026-01-08

**Summary:**

This paper introduces CoCoMix, a novel pretraining framework that augments standard next-token prediction with an auxiliary objective of predicting and interleaving “continuous concepts” from a sparse autoencoder. Reviewers find the idea original, well-presented, and impactful. The core merits are improved sample efficiency, a new paradigm for integrating interpretable concepts into pretraining, and demonstrated steerability. However, key concerns remain regarding the depth of validation for the concept-steering claims, the limited scale of models tested, the modest downstream gains, and the lack of statistical significance analysis and discussion of scaling challenges.

**Reviewer Concerns:**

The authors’ rebuttal addressed several clarifications: confirming PPL comparability, providing code, explaining concept selection/prediction, and detailing compute cost.

**Reviewer Scores:**

Reviewer SsAp (Initial: 8): This reviewer was positive, with minor concerns about hyperparameters. The satisfactory reply would likely maintain their score at 8. Reviewer XEZ8 (Initial: 6): This reviewer had significant methodological and validation concerns. The rebuttal and subsequent exchange highlighted that core issues (concept validation, scaling discussion, statistical evidence) were not substantively addressed. Their score would likely decrease, moving to 5 (borderline) or lower, reflecting increased doubt about the paper's conclusiveness. Reviewer LpV1 (Initial: 8): Most of their detailed questions were answered satisfactorily. While concern about model scale remains, they explicitly maintained support. Their score would likely stay at 8.

---

### Decision · Program_Chairs · 2026-01-26

Accept (Poster)